# Enhanced Detection of SARS-CoV-2 Using Platinum-Decorated Poly(2-vinylpyridine) Nanoparticle-Based Lateral Flow Immunoassay

**DOI:** 10.3390/biomedicines13122993

**Published:** 2025-12-06

**Authors:** Yayoi Kimura, Yasushi Enomoto, Yasufumi Matsumura, Kazuo Horikawa, Hideaki Kato, Atsushi Goto, Kei Miyakawa, Akihide Ryo

**Affiliations:** 1Advanced Medical Research Center, Yokohama City University, Yokohama 236-0004, Japan; 2Emerging Infectious Diseases Research Center, Yokohama City University Hospital, Yokohama 236-0004, Japan; 3New Materials R&D Lab., Advanced Materials Laboratories, R&D Div., Nippon Steel Chemical & Material Co., Ltd., Chiba 292-0835, Japan; 4Infection Prevention and Control Department, Yokohama City University Hospital, Yokohama 236-0004, Japan; 5Department of Public Health, Yokohama City University School of Medicine, Yokohama 236-0004, Japan; 6Department of Microbiology, Yokohama City University School of Medicine, Yokohama 236-0004, Japan; 7Influenza Research Center, National Institute of Infectious Diseases, Japan Institute for Health Security, Tokyo 208-0011, Japan; 8Department of Bioinformatics and Integrative Omics, National Institute of Infectious Diseases, Japan Institute for Health Security, Tokyo 208-0011, Japan

**Keywords:** human nasopharyngeal swab, immunochromatographic tests, lateral flow immunoassay, SARS-CoV-2

## Abstract

**Background:** Rapid and high-throughput diagnostic methods are essential for controlling the spread of infectious diseases, such as severe acute respiratory syndrome coronavirus 2 (SARS-CoV-2). Lateral flow immunoassay (LFIA) strips provide a cost-effective and user-friendly platform for point-of-care testing. However, the sensitivity of conventional LFIA kits is often limited by the performance of their detection probes. This study reports a highly sensitive LFIA strip for detecting the SARS-CoV-2 nucleocapsid (NP) protein using platinum-decorated poly(2-vinylpyridine) nanoparticles (Pt-P2VPs) as probes. **Methods:** Monoclonal antibodies against SARS-CoV-2 NP were conjugated with Pt-P2VPs and incorporated into LFIA strips. The test line was coated with anti–SARS-CoV-2 NP monoclonal antibody, and the control line with goat anti-mouse IgG. Recombinant proteins, viral strains, and nasopharyngeal swab specimens from patients were used to evaluate assay performance, with reverse transcription polymerase chain reaction (RT-PCR) as the reference standard. Diagnostic accuracy was assessed using nonparametric statistical tests. **Results:** Pt-P2VP-based LFIA strips enabled sensitive detection of recombinant NP and inactivated SARS-CoV-2, with minimal cross-reactivity. In 200 clinical specimens (100 PCR-negative and 100 PCR-positive), the assay achieved 74% sensitivity and 100% specificity, with strong correlation to viral RNA load. Compared with conventional LFIA kits, Pt-P2VP strips demonstrated superior sensitivity at lower viral loads. **Conclusions:** Pt-P2VPs represent a promising probe material for enhancing LFIA performance and may facilitate the development of rapid, sensitive, and scalable immunoassays for infectious disease diagnostics in biomedical applications.

## 1. Introduction

Highly transmissible respiratory infections, such as coronavirus disease 2019 (COVID-19) caused by severe acute respiratory syndrome coronavirus 2 (SARS-CoV-2), pose major global health challenges due to rapid person-to-person transmission and widespread dissemination [1,2,3]. Early and accurate detection of infected individuals is crucial for effective clinical management and containment of the disease. Moreover, even with the global rollout of COVID-19 vaccines, breakthrough infections and the emergence of immune-evasive variants continue to drive community transmission. Therefore, early and accurate detection is still a fundamental component of effective disease control strategies. Nucleic acid amplification tests, including real-time reverse transcription polymerase chain reaction (RT-PCR) and reverse transcription loop-mediated isothermal amplification, are widely recommended for their high sensitivity and specificity [4,5,6]. In parallel, novel diagnostic approaches such as CRISPR-based assays and microfluidic platforms have been explored [7,8]. However, these methods require specialized equipment, trained personnel, lengthy procedures, and notable costs, which limit their applicability in point-of-care settings. This underscores the ongoing need for rapid and sensitive diagnostic tools, particularly for early-stage detection and large-scale population screening.

Antigen-based assays, such as enzyme immunoassay (EIA), particle agglutination assay (PA), and Western blotting, provide alternative diagnostic approaches by detecting viral proteins [9,10]. While EIA offers high sensitivity, it requires complex and costly instrumentation. PA and Western blotting are less equipment-dependent but are time-intensive and unsuitable for rapid diagnostics. In contrast, lateral flow immunoassay (LFIA) strips have emerged as practical tools for rapid, cost-effective, and user-friendly biomarker detection in diverse biological fluids [10,11,12]. The results of LFIAs can be read visually without the need for any additional equipment or instrumentation. They have gained significant attention due to their simplicity, portability, and potential for point-of-care applications, and have been applied to various diagnostic targets, including SARS-CoV-2. However, conventional LFIA strips typically rely on colorimetric detection using colloidal gold nanoparticles, which are favored for their optical properties and stability, but these systems often suffer from limited sensitivity and high false-negative rates. Although the specificity of SARS-CoV-2 antigen-based LFIAs has consistently been reported as high, their sensitivity varies considerably among commercial products, with some demonstrating moderate to excellent performance, while others show suboptimal results, raising concerns about their reliability in clinical and screening settings [10].

Recent advancements in nanotechnology have introduced metal nanoparticle–latex nanocomposites, such as platinum-decorated poly(2-vinylpyridine) nanoparticles (Pt-P2VPs), which can enhance colorimetric signals and improve detection sensitivity [13]. Despite these advances, the clinical performance of LFIA strips incorporating optimized monoclonal antibodies (mAbs) with Pt-P2VP probes remains underexplored. Nasopharyngeal swab (NPS) specimens remain the most widely used sample type for SARS-CoV-2 detection because of their non-invasive nature and suitability for large-scale screening.

In this study, we developed highly sensitive LFIA strips by incorporating optimized anti–SARS-CoV-2 nucleocapsid protein (NP) mAbs and Pt-P2VP probes, hereafter referred to as Pt-P2VP-based LFIA strips [13,14]. To demonstrate their high sensitivity, we evaluated their diagnostic performance using recombinant antigens, viral particles, and NPS specimens from patients with confirmed SARS-CoV-2 infection. Furthermore, we compared their sensitivity and specificity with those of conventional diagnostic kits to demonstrate the clinical advantages of this approach. Our findings may highlight the potential of Pt-P2VPs as versatile probes for the rapid, reliable, and scalable detection of SARS-CoV-2 and other infectious agents, particularly in resource-limited or decentralized settings.

## 2. Materials and Methods

### 2.1. Conjugation of Anti-SARS-CoV-2 NP mAb with Pt-P2VP Nanoparticles

Anti–SARS-CoV-2 NP monoclonal antibodies (mAbs) were conjugated with Pt-P2VPs. Briefly, 294 μL of 1.0% (*w*/*v*) aqueous Pt-P2VP dispersion (430 nm diameter) was mixed with 11.8 μL of 5 mg/mL anti-NP mAb (cat. 02126-67, Appendix A) in 2046 μL of 50 mM HEPES buffer (pH 7.0). The mixture was incubated at 25 °C for 2 h. Subsequently, 588 μL of 5.0% (*w*/*v*) aqueous sodium casein solution was added and briefly sonicated. The suspension was then incubated at 30 °C for 1 h, followed by centrifugation at 2500× *g* for 3 min. After discarding the supernatant, the resulting mAb-conjugated Pt-P2VPs were washed three times with 5 mM Tris-HCl (pH 8.5) and resuspended in 28 mL of storage buffer. Finally, the suspension was applied to a conjugate pad and dried at 45 °C for 90 min.

### 2.2. Assembly of LFIA Strips

LFIA strips (60 mm × 4 mm) were assembled with the following components in sequence: a sample pad (15 mm × 4 mm), a conjugate pad (10 mm × 5 mm), a nitrocellulose membrane (25 mm × 4 mm), and an absorbent pad (20 mm × 4 mm). Adjacent layers were overlapped by 2–5 mm to ensure continuous fluid flow (Figure 1a).

The nitrocellulose membrane was printed with an anti-SARS-CoV-2 NP mAb (cat. 02128-67, Appendix A) at 28 mm from the bottom as the test line and with a goat anti-mouse IgG antibody at 35 mm as the control line. Quality control during the manufacturing process included OD measurement and concentration adjustment of the mAb-conjugated Pt-P2VP suspension, visual inspection of the membrane antibody coating for line position and stability, and alignment of the pad during strip assembly.

### 2.3. Preparation of Recombinant NP, Viral Samples, and Clinical Specimens

Recombinant SARS-CoV-2 NPs with N-terminal deletions (cat. 38157-96, Appendix A) were used in this study. Viral strains and NPS specimens were prepared as previously described [14]. All samples were stored at −80 °C until use and were inactivated by adding NP-40 to a final concentration of 0.5% (*v*/*v*) immediately prior to each immunoassay. The SARS-CoV-2 and other viral strains used in this study are listed in Appendix A.

Clinical NPS specimens were collected from patients who visited Yokohama City University Hospital between July 2021 and July 2022. The study protocol was approved by the Clinical Ethics Committee of Yokohama City University Hospital (approval no. B200800106), and written informed consent was waived because the study employed an opt-out approach in accordance with the committee’s guidelines. The study was conducted in accordance with the Declaration of Helsinki, and all data were anonymized before analysis.

NPS specimens, pre-diluted in 3 mL of transport medium and previously diagnosed as SARS-CoV-2 positive or negative by administrative inspection [15], were centrifuged at 4 °C for 30 min. Supernatants were stored at −80 °C until use. For viral RNA quantification, 140 μL of diluted NPS specimens were processed using the QIAamp Viral RNA Mini Kit. One-step RT-PCR was performed using the CFX96 Touch Real-Time PCR Detection Systemwith the NIID-N2 primer set. For LFIA testing, 20 μL of diluted NPS specimens were further diluted 10- or 20-fold in the respective extraction buffer, and 50 μL or 100 μL was applied per kit, respectively (Figure 1b). Threshold cycle (Ct) values of diluted samples were adjusted based on viral copy numbers.

### 2.4. Quantification and Statistical Analysis

Statistical analyses were conducted using GraphPad Prism 8 software and EZR software (version 1.61). Detailed statistical information is provided in the figure legends.

## 3. Results

### 3.1. Development of LFIA Strips for SARS-CoV-2 Detection

In this study, we employed Pt-P2VP nanoparticles as probes for LFIA in combination with an optimized pair of mAbs that specifically recognize SARS-CoV-2 NP. These mAbs do not cross-react with other human coronaviruses or respiratory viruses [14]. To evaluate the antigen detection performance of the LFIA strips under the consistent experimental protocol, we conducted assays using recombinant SARS-CoV-2 NP at concentrations of 0, 0.2, and 2.0 ng/mL (*n* = 15 per group). The repeatability of the qualitative detection assay was evaluated by calculating the coefficient of variation, which was 18.4% for the 0.2 ng/mL group and 17.6% for the 2.0 ng/mL group (Figure 2a). The statistical analysis revealed a significant difference among the three concentration groups (*p* < 0.01), indicating excellent discriminatory capability of the assay.

A dose-dependent response was observed across a range of recombinant NP concentrations (0–2.0 ng/mL, *n* = 3 per group), with absorbance values correlating linearly with antigen levels, consistent with the colorimetric detection results (Figure 2b). These findings indicate that the LFIA strips enable semi-quantitative detection of SARS-CoV-2 NP. The colorimetric detection threshold was determined to be 0.1 ng/mL of recombinant NP, corresponding to an absorbance threshold of 7 mAbs at 10 min post-application (Figure 2c).

To evaluate viral detection capability, inactivated virions from major SARS-CoV-2 Omicron strains were tested. A dose-dependent linear response was observed in colorimetric measurements across 0–2.0 × 10^7^ copies/mL for SARS-CoV-2 Omicron_XBB.1 (TY41-795) (Figure 2d), consistent with results obtained using recombinant NP. The strips reliably detected viral loads ranging from 1.0 × 10^1^ to 1.3 × 10^3^ TCID_50_/mL for five inactivated virions listed in Appendix A (*n* = 3 per virus).

Sequence analysis of emerging Omicron subvariants (JN.1, KP.3, and XEC) confirmed the conservation of the NP epitopes targeted by the mAbs, supporting the continued applicability of the LFIA platform. Cross-reactivity testing with other human coronaviruses and common respiratory viruses, including influenza A, influenza B, human rhinovirus, and respiratory syncytial virus, showed no detectable cross-reactivity at concentrations ranging from 1.1 × 10^3^ to 6.3 × 10^4^ TCID_50_/mL (*n* = 3 per virus) (Appendix A).

These results demonstrate that Pt-P2VP-based LFIA strips provide stable and sensitive detection of SARS-CoV-2, with minimal cross-reactivity with other respiratory pathogens.

### 3.2. Clinical Performance of LFIA Strips for SARS-CoV-2 Detection

To evaluate the clinical performance of Pt-P2VP-based LFIA strips for SARS-CoV-2 detection, we tested NPS specimens from patients diagnosed with SARS-CoV-2 by RT-PCR. A total of 100 PCR-negative and 100 PCR-positive specimens were retrospectively selected (Appendix A). Among the PCR-positive specimens, 74 were visually positive on LFIA strips, whereas all PCR-negative specimens tested negative. The resulting sensitivity, specificity, positive predictive value, and negative predictive value were 74.0% (95% confidence interval (CI): 64.3–82.3%), 100% (95% CI: 94.6–100%), 100% (95% CI: 92.8–100%), and 74.9% (95% CI: 71.2–86.1%), respectively (Table 1).

On stratifying the PCR-positive samples by Ct values, the LFIA strips showed 100% sensitivity for Ct < 27, 60.0% for Ct 27–30, and 0% for Ct > 30 (Table 2).

Absorbance measurements correlated strongly with SARS-CoV-2 RNA copy numbers (r = 0.9495), and the colorimetric detection threshold was 7 mAbs, consistent with results from recombinant NP assays (Appendix A).

Additionally, the sensitivity of Pt-P2VP-based LFIA strips was compared with two conventional LFIA strips that employ colorimetric detection using colored latex particles and colloidal gold nanoparticles, using 30 PCR-positive NPS specimens with varying Ct values (Appendix A). In the Ct value range of 27–30, the sensitivity of the Pt-P2VP-based LFIA strips was 75.0% (50.9–91.3), compared with 25.0% (8.7–49.1) for Kit A and 50.0% (27.2–72.8) for Kit B (Table 3). Therefore, the Pt-P2VP-based LFIA strips demonstrated greater sensitivity than conventional kits at lower viral loads, whereas no significant differences were observed at higher viral loads.

Conjugation with Pt-P2VP likely enhanced the detection of antigen–antibody complexes. These findings support the utility of Pt-P2VP-based LFIA strips for rapid and reliable SARS-CoV-2 antigen detection in clinical settings.

## 4. Discussion

RT-PCR remains the gold standard for diagnosing infectious diseases, such as SARS-CoV-2, in clinical practice. However, during pandemics, rapid, high-throughput, and accessible diagnostic tools are critical for mitigating transmission. LFIA strips represent a promising alternative due to their rapid turnaround, operational simplicity, and suitability for point-of-care testing. Despite these advantages, conventional LFIA strips are often limited by suboptimal sensitivity, primarily determined by the performance of their detection probes. Low-sensitivity tests, while scalable and cost-effective, may yield false-negative results, particularly in individuals with low viral loads, thereby facilitating further transmission. Enhancing detection sensitivity to levels comparable with enzyme-linked immunosorbent assays and chemiluminescent EIAs remains a key challenge for the broader clinical adoption of LFIA strips.

To address this limitation, Pt-P2VPs were developed as detection probes for LFIA platforms [13]. Their structural integrity, characterized by firmly anchored metal nanoparticles, monodispersity, and colloidal stability, contributes to long-term maintenance of strong colorimetric signals [13]. Pt-P2VP-based LFIA strips therefore offer practical advantages, including ease of manufacturing, stability, minimal operator dependency, and compatibility with digital readers and information systems. In this study, we developed LFIA strips by incorporating highly sensitive mAbs and Pt-P2VP probes for detecting the SARS-CoV-2 NP protein. As a clinical advantage, these strips demonstrated improved detection sensitivity compared with conventional kits, particularly at low viral concentrations. This enhancement is particularly significant in the context of viral shedding dynamics, which influence transmission risk and diagnostic timing [16]. A systematic review of 14 studies involving serial viral culture testing reported positive viral cultures from specimens collected 4 days before to 18 days after symptom onset [17]. Daily positivity rates declined between days 5 and 9, peaking at 44–50% from days –1 to 5, decreasing to 28% on day 7, 11% on day 9, and 0–8% on days 10–17. Notably, viral isolation occurred even before symptom onset, highlighting the need for frequent and sensitive screening to prevent outbreaks [16]. Pt-P2VP-based LFIA strips enable on-site testing with improved sensitivity, particularly for individuals with low viral loads during early or late infection, supporting their use in both clinical and community-based screening to reduce false-negative results and mitigate transmission risk.

Previous studies have demonstrated that Pt-P2VP probes improve LFIA sensitivity for influenza A (H1N1) antigen detection compared with conventional gold colloid probes [13]. Further improvements were reported by Le et al., who chemically bound Pt-P2VPs to superparamagnetic iron oxide nanoparticles, enabling magnetic separation for sample enrichment and purification [18]. This strategy reduced the visual limit of detection by 26-fold relative to standard gold nanoparticle-based LFIAs, highlighting the potential of Pt-P2VPs, especially when integrated with other advanced technologies, for developing highly sensitive and broadly applicable LFIA platforms.

This study has some limitations. First, it employed antibodies different from those used in conventional LFIA strips; direct comparisons using the same antibodies are warranted in future studies. Second, this was a retrospective analysis conducted under controlled laboratory conditions. Therefore, prospective, real-world evaluations across diverse populations and settings are needed to confirm clinical utility. Furthermore, the stability and ease of production of Pt-P2VP-based LFIA strips make them suitable for scalable deployment in resource-limited settings, thereby contributing to equitable access to reliable diagnostics during future public health emergencies. In this study, we employed a prototype LFIA strip that was created following actual manufacturing processes. The Pt-P2VP material used in the test line can be supplied consistently, which is ideal for large-scale production. These features support the potential for large-scale implementation of this platform, not only for SARS-CoV-2 detection but also for various diagnostic applications in clinical and point-of-care settings.

## 5. Conclusions

This study demonstrated that Pt-P2VP-based LFIA strips provide enhanced sensitivity for SARS-CoV-2 antigen detection, particularly at low viral loads, making them suitable for early-stage diagnosis and large-scale population screening. Pt-P2VPs represent a versatile and promising platform for improving LFIA performance and have the potential for broader application in screening tests for both infectious and noninfectious diseases in biomedical settings.

## Figures and Tables

**Figure 1 biomedicines-13-02993-f001:**
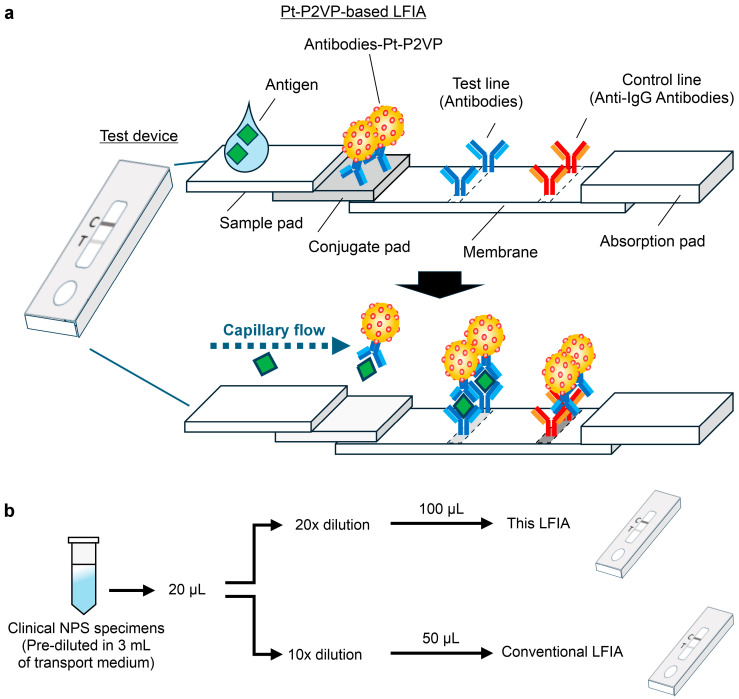
Overview and experimental procedure of lateral flow immunoassay (LFIA) strips for antigen detection. (**a**) Schematic representation of the LFIA strip incorporating the optimal pair of anti-severe acute respiratory syndrome coronavirus 2 (SARS-CoV-2) nucleocapsid protein (NP) monoclonal antibodies conjugated with platinum-decorated poly(2-vinylpyridine) nanocomposites (Pt-P2VPs). (**b**) Experimental workflow for the clinical performance evaluation using nasopharyngeal swab specimens tested by reverse transcription polymerase chain reaction. The specimens were standardized by dilution rate and applied volume to ensure a consistent viral load across tests. LFIA strips were tested using a custom extraction buffer (100 mM Tris-HCl (pH 7.5) buffer containing 150 mM NaCl, 1% (*v*/*v*) NP-40, and 0.2% (*w*/*v*) sodium caseinate), whereas conventional kits used their respective supplied buffers. Test lines were measured using an optical reader (model C10066-10) at 10 min (Pt-P2VP-based LFIA), 8 min (Kit A), and 15 min (Kit B), followed by colorimetric detection.

**Figure 2 biomedicines-13-02993-f002:**
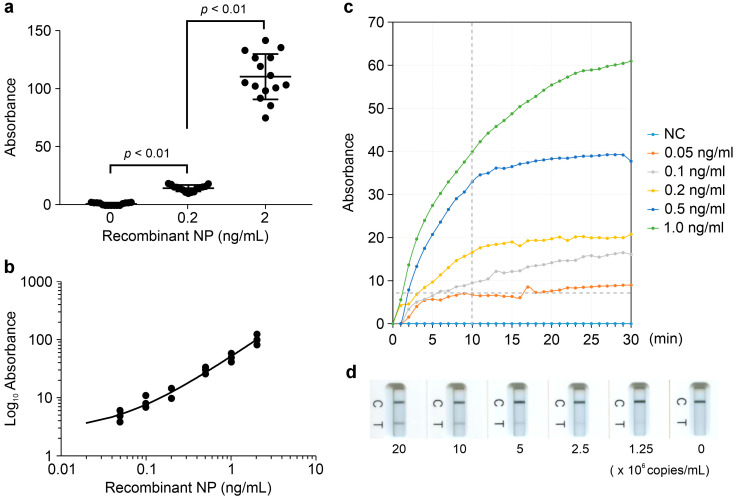
Analytical performance of Pt-P2VP-based LFIA strips. (**a**) Repeatability of LFIA strips with 0, 0.2, or 2.0 ng/mL recombinant NP under a consistent experimental protocol. Data are presented as median ± standard deviation (SD) (*n* = 15). Comparisons among the three groups were performed using the Kruskal–Wallis test and Dunn’s multiple comparison test; (**b**) Detection results across serial dilutions of recombinant NP (0–2.0 ng/mL) for evaluation of a linear correlation between NP concentration and absorbance values. (**c**) Time-dependent changes in absorbance at each NP concentration for determination of the colorimetric detection threshold of recombinant NP. NC indicates the negative control. “Recombinant NP” refers to the recombinant NP antigen of the human coronavirus. (**d**) Detection results across serial dilutions of inactivated virions using the SARS-CoV-2 Omicron_XBB.1 (TY41-795) strain (0–2.0 × 10^7^ copies/mL) for evaluation of viral detection capability.

**Table 1 biomedicines-13-02993-t001:** Sensitivity and specificity of colorimetric detection using lateral flow immunoassay (LFIA) strips against 100 PCR-positive and 100 PCR-negative clinical NPS specimens.

		RT-PCR	
		Positive	Negative
LFIA strips	Positive	74	0
	Negative	26	100
Sensitivity (%) (95% CI) *		74.0 (64.3–82.3)	
Specificity (%) (95% CI) *		100 (94.6–100)	
Positive Predictive Value (%) (95% CI) *		100 (92.8–100)	
Negative Predictive Value (%) (95% CI) *		74.9 (71.2–86.1)	

* The exact confidence intervals for sensitivity and specificity were calculated using the Clopper–Pearson method.

**Table 2 biomedicines-13-02993-t002:** Sensitivity stratified by Ct value ranges (<27, 27–30, >30) among 100 PCR-positive specimens.

Adjusted Ct Values	LFIA Strips		Sensitivity (%) (95% CI) *
	Positive	Negative	
<27	59	0	100.0 (91.1–100)
27–30	15	10	60.0 (38.7–78.9)
>30	0	16	0.0 (0.0–28.7)

* The exact confidence intervals for sensitivity and specificity were calculated using the Clopper–Pearson method.

**Table 3 biomedicines-13-02993-t003:** Comparative sensitivity of Pt-P2VP-based LFIA strips and two conventional LFIA kits using 30 PCR-positive clinical NPS specimens categorized by Ct value ranges.

Adjusted Ct Values	Pt-P2VP-Based LFIA Strips		Sensitivity (%)	Kit A		Sensitivity (%)	Kit B		Sensitivity (%)
	Positive	Negative		Positive	Negative		Positive	Negative	
<27	10	0	100.0	10	0	100.0	10	0	100.0
27–31	15	5	75.0	5	15	25.0	10	10	50.0
Total	25	5	83.3	15	15	50.0	20	10	66.7

## Data Availability

The original contributions presented in this study are included in the article/Appendix A. Further inquiries can be directed to the corresponding author.

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
