# Peer review of "Biomedicines2025, 13(12), 2993;https://doi.org/10.3390/biomedicines13122993"

_biomedicines, 2025, doi:10.3390/biomedicines13122993_

Round 1

Reviewer 1 Report

Comments and Suggestions for Authors

Enhanced Detection of SARS-CoV-2 Using Platinum-Decorated Poly(2-vinylpyridine) Nanoparticle–Based Lateral Flow Immunoassay

Ryo et al., present a very interesting and yet timely article where they This study reports a highly sensitive LFIA strip for detecting the SARS-CoV-2 nucleocapsid protein using platinum-decorated poly(2-vinylpyridine) nanoparticles (Pt-P2VPs) as probes

demonstrating that Pt-P2VP-based LFIA strips provide enhanced sensitivity for SARS-CoV-2 antigen detection, particularly at low viral loads, making them suitable for early-stage diagnosis and large-scale population screening. It’s well written and concise. However, despite these merits this manuscript has certain flaws or rather clarifications or limitations and addressing those criticisms might put this manuscript in a pretty good spot for consideration for publication. For the starters, intro doesn’t cover the background and thus not comprehensible. I’d mention novel Covid vaccines and approaches, either in intro for improving background or discussion the very least 

Ref 8-10, I’d cite the attached since the intro has been sparingly referenced. 

Section 2.1 ug volumes are mentioned but not the concentrations? Why ? Explain.

All the catalogue numbers mentioned in the main text, maybe put it in a table and add at the end of the manuscript or supplementary information. It’s not necessary in the main text maybe.

Line 147, volumes are mentioned but not the concentration, please provide concentrations in addition to volumes wherever possible. 

Line 161 check line and spacing. 

I’d improve figure legend for figure 2. It should reflect the content of the figure in figure legend in addition to the main text.

Please check grammar, spelling and punctuation. Avoid complex sentences and awkward phrasing. 

Check all the references for doi, page and volume numbers. 

Best of luck and keep up the good work,

Cheers!

Comments on the Quality of English Language

Enhanced Detection of SARS-CoV-2 Using Platinum-Decorated Poly(2-vinylpyridine) Nanoparticle–Based Lateral Flow Immunoassay

Ryo et al., present a very interesting and yet timely article where they This study reports a highly sensitive LFIA strip for detecting the SARS-CoV-2 nucleocapsid protein using platinum-decorated poly(2-vinylpyridine) nanoparticles (Pt-P2VPs) as probes

demonstrating that Pt-P2VP-based LFIA strips provide enhanced sensitivity for SARS-CoV-2 antigen detection, particularly at low viral loads, making them suitable for early-stage diagnosis and large-scale population screening. It’s well written and concise. However, despite these merits this manuscript has certain flaws or rather clarifications or limitations and addressing those criticisms might put this manuscript in a pretty good spot for consideration for publication. For the starters, intro doesn’t cover the background and thus not comprehensible. I’d mention novel Covid vaccines and approaches, either in intro for improving background or discussion the very least

Ref 8-10, I’d cite the attached since the intro has been sparingly referenced. 

Section 2.1 ug volumes are mentioned but not the concentrations? Why ? Explain.

All the catalogue numbers mentioned in the main text, maybe put it in a table and add at the end of the manuscript or supplementary information. It’s not necessary in the main text maybe.

Line 147, volumes are mentioned but not the concentration, please provide concentrations in addition to volumes wherever possible. 

Line 161 check line and spacing. 

I’d improve figure legend for figure 2. It should reflect the content of the figure in figure legend in addition to the main text.

Please check grammar, spelling and punctuation. Avoid complex sentences and awkward phrasing. 

Check all the references for doi, page and volume numbers. 

Best of luck and keep up the good work,

Cheers!

Author Response

Ryo et al., present a very interesting and yet timely article where they This study reports a highly sensitive LFIA strip for detecting the SARS-CoV-2 nucleocapsid protein using platinum-decorated poly(2-vinylpyridine) nanoparticles (Pt-P2VPs) as probes.

We appreciate the reviewer’s constructive suggestions. We have carefully considered the comments and revised the manuscript accordingly.

  1. demonstrating that Pt-P2VP-based LFIA strips provide enhanced sensitivity for SARS-CoV-2 antigen detection, particularly at low viral loads, making them suitable for early-stage diagnosis and large-scale population screening. It’s well written and concise. However, despite these merits this manuscript has certain flaws or rather clarifications or limitations and addressing those criticisms might put this manuscript in a pretty good spot for consideration for publication. For the starters, intro doesn’t cover the background and thus not comprehensible. I’d mention novel Covid vaccines and approaches, either in intro for improving background or discussion the very least.

Response: Thank you for your constructive feedback. In response to your suggestion, we have revised the Introduction section to better contextualize our study within the broader landscape of COVID-19 diagnostics. Specifically, we have added new sentences and references 7 (Sahel DK et al.) and 8 (Tarin EA et al.) to emphasize the relevance of our work in light of ongoing vaccination efforts and emerging diagnostic technologies. The previous reference 7 and subsequent references have been renumbered to reference 9 and above accordingly.

  1. Ref 8-10, I’d cite the attached since the intro has been sparingly referenced. 

Response: Thank you for the helpful suggestion. In response, we have cited the recommended literature [8–10], which provides comprehensive background information on LFIAs, and revised the relevant sections.

  1. Section 2.1 ug volumes are mentioned but not the concentrations? Why ? Explain.

Response: Thank you for pointing this out. In the revised manuscript, we have clarified the concentration of the anti–SARS-CoV-2 NP monoclonal antibody by specifying both the volume (11.8 μL) and the concentration (5 mg/mL).

  1. All the catalogue numbers mentioned in the main text, maybe put it in a table and add at the end of the manuscript or supplementary information. It’s not necessary in the main text maybe.

Response: Thank you for your valuable comment. To improve readability, we have removed all supplier information and selected catalogue numbers from the main text and compiled them into a table in the supplementary information.

  1. Line 147, volumes are mentioned but not the concentration, please provide concentrations in addition to volumes wherever possible. 

Response: Thank you for your comment. We have reviewed Line 147 and found that it does not contain any volume-related descriptions. If your concern pertains to the viral concentration in clinical specimens, we would like to clarify that the concentration is inherently variable across samples. Therefore, we have provided the sample volumes in the main text and summarized the corresponding viral copy numbers in Supplementary Table S2.
If your comment refers to a different aspect, we would appreciate further clarification so that we can address it appropriately.

  1. Line 161 check line and spacing. 

Response: Thank you for your valuable comment. As per your suggestion, we have revised the formatting at Line 161 by adding a line break and adjusting the spacing between the text and the figure to improve clarity and layout.

  1. I’d improve figure legend for figure 2. It should reflect the content of the figure in figure legend in addition to the main text.

Response: Thank you for your valuable comment. As per your suggestion, we have revised the legend for Figure 2 to better reflect the experimental purpose. The updated legend now includes information on the evaluation of linear correlation, detection threshold, and viral detection using the Omicron_XBB.1 strain.

  1. Please check grammar, spelling and punctuation. Avoid complex sentences and awkward phrasing. 

Response: Thank you for your valuable comment. The manuscript was thoroughly reviewed by another native speaker, and the grammatical and typographical errors, including those specified in the reviewer's comment, were rectified in the revised manuscript.

Reviewer 2 Report

Comments and Suggestions for Authors

This manuscript provided a significant detective LFIA strip for detection of clinical samples from infected SARS-CoV-2 patients, and showed the some quality evaluation about this production. It could be significant for detection of liquid samples form infected patients by current epidemic strains of SARS-CoV-2, even if there were some similar productors for some time. The reviewer suggested that it could be published after some minor improvement as below:

1: To add description about the preparation method of this productio in large scale;

2: To list and make their explanation about the items of quality control of this production. 

Author Response

This manuscript provided a significant detective LFIA strip for detection of clinical samples from infected SARS-CoV-2 patients, and showed the some quality evaluation about this production. It could be significant for detection of liquid samples form infected patients by current epidemic strains of SARS-CoV-2, even if there were some similar productors for some time. The reviewer suggested that it could be published after some minor improvement as below:

Response: We appreciate the reviewer’s valuable suggestions. We have carefully considered the comments and revised the manuscript accordingly.

1: To add description about the preparation method of this productio in large scale;

Response: Thank you for your valuable comment. In this study, we used a prototype that was developed in accordance with actual manufacturing processes. The Pt-P2VP material used in the LFIA strips is consistently available and suitable for scalable production. These features support the potential for large-scale implementation of this platform, not only for SARS-CoV-2 detection but also for broader diagnostic applications in clinical and point-of-care settings. This point has been addressed by adding a corresponding statement at the end of the Discussion section.

2: To list and make their explanation about the items of quality control of this production. 

Response: Thank you for your valuable suggestion. The quality control items related to the manufacturing process and product performance evaluation are as follows:

Quality control during the manufacturing process:

  1. Measurement of optical density (OD) and concentration adjustment of the mAb-conjugated Pt-P2VPs suspension.
  2. Visual inspection of membrane antibody coating for line position and stability.
  3. Visual inspection of pad alignment during LFIA strip assembly.

Product performance tests:

  1. Simultaneous reproducibility test using recombinant NP at 0, 0.2, and 2.0 ng/mL (n = 5).
  2. Repeatability test performed three times under the same conditions.

The results of the product performance tests are presented in Figure 2a. Accordingly, we have added a description of the quality control items related to the manufacturing process at the end of Section 2.2 in the revised manuscript.
